# Time-travelling pathogens and their risk to ecological communities

**Giovanni Strona** [1,2]*, **Corey J. A. Bradshaw** [3,4], **Pedro Cardoso**[5], **Nicholas J. Gotelli**[6], **Frédéric Guillaume**[2], **Federica Manca**[2], **Ville Mustonen**[2,7], **Luis Zaman**[8,9]

1 European Commission, Joint Research Centre, Directorate D–Sustainable Resources, Ispra, Italy, 2 Faculty of Biological and Environmental Sciences, Organismal and Evolutionary Biology Research Programme, University of Helsinki, Helsinki, Finland, 3 Global Ecology | *Partuyarta Ngadluku Wardli Kuu*, College of Science and Engineering, Flinders University, Adelaide, South Australia, Australia, 4 ARC Centre of Excellence for Australian Biodiversity and Heritage, Wollongong, Australia, 5 Laboratory for Integrative Biodiversity Research—LIBRe, Finnish Museum of Natural History Luomus, University of Helsinki, Helsinki, Finland, 6 Department of Biology, University of Vermont, Burlington, Vermont, United States of America, 7 Institute of Biotechnology, Department of Computer Science, University of Helsinki, Helsinki, Finland, 8 Department of Ecology and Evolutionary Biology, University of Michigan, Ann Arbor, Michigan, United States of America, 9 Center for the Study of Complex Systems, University of Michigan, Ann Arbor, Michigan, United States of America

* goblinshrimp@gmail.com

**Data Availability Statement:** We processed Avida output using custom Python scripts, and did the analyses on the resulting data using R. All the scripts necessary to replicate the analyses as well

## Abstract

Permafrost thawing and the potential 'lab leak' of ancient microorganisms generate risks of biological invasions for today's ecological communities, including threats to human health via exposure to emergent pathogens. Whether and how such 'time-travelling' invaders could establish in modern communities is unclear, and existing data are too scarce to test hypotheses. To quantify the risks of time-travelling invasions, we isolated digital virus-like pathogens from the past records of coevolved artificial life communities and studied their simulated invasion into future states of the community. We then investigated how invasions affected diversity of the free-living bacteria-like organisms (i.e., hosts) in recipient communities compared to controls where no invasion occurred (and control invasions of contemporary pathogens). Invading pathogens could often survive and continue evolving, and in a few cases (3.1%) became exceptionally dominant in the invaded community. Even so, invaders often had negligible effects on the invaded community composition; however, in a few, highly unpredictable cases (1.1%), invaders precipitated either substantial losses (up to -32%) or gains (up to +12%) in the total richness of free-living species compared to controls. Given the sheer abundance of ancient microorganisms regularly released into modern communities, such a low probability of outbreak events still presents substantial risks. Our findings therefore suggest that unpredictable threats so far confined to science fiction and conjecture could in fact be powerful drivers of ecological change.

## Author summary

The idea that ancient pathogens trapped in ice or hidden in remote laboratory facilities could break free—usually with catastrophic consequences for human beings—has been a

as the Avida configuration files and the processed Avida output are available at (https://github.com/giovannistrona/tti).

**Funding:** GS, PC, VM and LZ where partly supported by a "HiLIFE BIORESLIENCE seed grant" from the University of Helsinki (https://www.helsinki.fi/en/hilife-helsinki-institute-life-science/research/grand-challenges/understanding-biological-resilience-bioresilience). The funders had no role in study design, data collection and analysis, decision to publish, or preparation of the manuscript.

**Competing interests:** The authors have declared that no competing interests exist.

fruitful source of inspiration for generations of science fiction novelists and screenwriters. However, the unprecedented rates of melting of glaciers and permafrost are now giving many types of ice-dormant microorganisms concrete opportunities to re-emerge, bringing to the fore questions about their potential. Yet, the scientific debate on the topic has been dominated by speculation, due to the challenges in collecting appropriate data or designing experiments to elaborate and test hypotheses. For the first time, we provide an extensive exploration of the ecological risk posed to modern ecological communities by these 'time-travelling' pathogens by taking advantage of the flexibility and realism of *in silico* simulations. We found that invading pathogens could often survive, evolve and, in a few cases, become exceptionally persistent and dominant in the invaded community, causing either substantial losses or gains in the total richness of free-living species. Our findings therefore suggest that unpredictable threats so far confined to science fiction and conjecture could be powerful drivers of ecological change.

## Introduction

Biological invasions constitute a large potential threat to biodiversity [1–5] and human societies in the form of novel, emergent pathogens, as well as massive economic costs [6,7]. When a species is moved passively or actively from its range to a new locality with different environmental and ecological conditions, the consequences for native communities are unpredictable [8–10], but are often exceptionally severe when they succeed [11]. When catastrophic invasions do occur, it is most often a consequence of the lack of co-adaptation between the invader and native species [12]. Sharing a co-evolutionary history requires prolonged temporal and spatial overlap of species occurrence, but there is no such shared history when an alien species is first introduced into a naïve community. If until now most studies on invading species have examined the spatial aspect of the travellers, the paradigm is shifting, and time travellers might pose a future risk for biodiversity given the current acceleration of climate change and its consequences. In this sense, 'first time' can refer to a sudden event triggering exposure to an erstwhile temporally isolated organism.

Science fiction is rife with conjecture regarding the awakening of long-dormant organisms with unpredictable, yet serious outcomes for modern ecological systems and human societies. However, there are several environmental settings where there is a non-negligible probability of long-dormant organisms being exposed to modern communities. In particular, unprecedented rates of melting of glaciers and permafrost [13,14] are now giving many types of ice-dormant microorganisms the opportunity to re-emerge [15–21]. There is also potential for ancient (or de-extinct) microorganisms to leak from laboratory facilities [22]. Such catastrophic events, while individually unlikely to succeed, potentially represent massive threats to extant ecosystems given the sheer frequency of exposure. However, the risk has so far remained unquantifiable, calling for a comprehensive investigation of the likely eco-evolutionary mechanisms underlying the process of ancient invaders negatively affecting modern communities.

The implications of such emergent pathogens on human health have been discussed [23], and there have been some short time-scale laboratory experiments testing the interactions between 'ancestral' strains of bacteria and phylogenetically younger phages [24]. However, the potential ecological relevance of time-travelling invaders remains largely unexplored, especially with respect to community-level repercussions. This knowledge gap is primarily due to the challenges of collecting relevant data or setting up adequate experiments involving more

than few species (an issue that also partly affects the study of spatial invasions [25]). In this context, artificial life simulations—where entire communities of simple organisms can be studied at both ecological and evolutionary timescales—offer a powerful tool to circumvent these challenges and obtain heretofore unexplored insight.

We constructed a large set of artificial evolution experiments where digital virus-like pathogens from the past invade communities of bacteria-like hosts and their contemporary pathogens. For this, we used *Avida*, an artificial life system simulating *in silico* evolution of complex communities of digital micro-organisms ecologically similar to bacteriophage viruses [26]. The *Avida* world consists of a bi-dimensional grid where sessile digital organisms interact with the environment by doing logical operations (i.e., 'tasks'), which in turn define their phenotype in functional terms (i.e., phenotypes are defined by the set of logical operations organisms can compute). In the simulated world, organisms compete for CPU cycles—the source of energy permitting them to reproduce—and for space. Time in *Avida* is measured in *updates*, where during each update organisms in the population have each executed an average of 30 single instructions (*i.e.*, 30 CPU cycles). To reproduce, organisms must copy their genetic code, consisting of simple computer instructions, into another location in memory before spawning a new *nearly* identical offspring organism (like what some classes of computer viruses do). While copying themselves, organisms can make errors. Most of the errors are deleterious, resulting in organisms that cannot reproduce. Yet, in some cases, the errors do not affect the new organism's competitive ability, or can even provide it with a competitive advantage over other organisms. In such cases, the new genotype might engender a higher chance of becoming fixed in the population.

In this way, communities of digital organisms become complex through processes of 'natural' selection, starting from a single, viable ancestral genotype. In *Avida*, organisms can have free-living and pathogenic lifestyles (like those of bacteriophage viruses). These lifestyles are distinct, meaning that free-living species cannot become (or evolve into) pathogens and vice versa. Avidian hosts and pathogens are similar in their construction and 'biology', but with the difference that the latter cannot draw energy directly from the environment. Instead, pathogens in *Avida* use CPU cycles originally allocated to their hosts, thus reducing their host's reproductive ability. Pathogens can only survive on suitable hosts and cannot move from one host to another. Therefore, novel infections happen with the transmission of a pathogenic propagule from an infected to a susceptible host (with propagules randomly allocated to grid cells and surviving only if occurring in a grid cell occupied by a suitable host). A given host can be infected by only one pathogen at a time. Pathogens can infect hosts based on a task-matching mechanisms (*i.e.*, for infection to be successful, the infecting pathogen must be able to do at least one of the logical tasks done by the potential host). Thus, acquiring the ability to do novel tasks (or, in general, tasks done uncommonly by pathogens) can confer an adaptive resistance to hosts, while pathogens can expand their host range and/or increase their prevalence into Avidian communities by evolving the ability to do 'popular' tasks. Because the possible logical tasks have different complexity, meaning that they require a different theoretical minimum number of basic code building blocks to be done, and hence pose different evolutionary challenges, host-pathogen dynamics give rise to a co-evolutionary arms race promoting the natural emergence of organismal complexity [27]. The bipartite antagonistic networks derived from these co-evolutionary processes are realistic and match those observed in real-world systems [28,29]. In our experiments, doing tasks requires using some specific quantity of a specific resource, and results in the 'excretion' in the environment of by-product resources according to a pre-defined configurable 'biochemistry' [26]. This means that the task done by the different organisms are both contingent on and affect the environmental conditions, which adds realism to the simulations (*e.g.*, by creating the potential for phenotypic plasticity in Avidians [30]).

Using *Avida*, we evolved *in silico* complex communities of digital organisms starting from a single ancestor under a broad range of environmental settings for many generations, keeping track of community changes over time. We then translocated pathogens from ancient communities to their respective modern analogues (Fig 1). We hypothesized different potential outcomes for these time-travelling invasions (see Methods for full description of and justification for hypotheses). On the one hand, because an organism's average fitness and complexity are expected to increase with time in *Avida* [31], we expect ancient pathogens to be more susceptible to competition from modern ones. Modern hosts might also have escaped from ancient pathogens during their co-evolutionary history, and possibly retain their evolved resistance, thereby challenging invaders to find susceptible hosts. However, we can envision contrary scenarios where modern hosts have lost this resistance if enough time has passed from their past co-existence with the invader (possibly leaving a 'vacated niche' that might be re-occupied by the pathogen if reintroduced [32]). In that case, invaders might have an advantage over modern pathogens actively involved in the ongoing host-pathogen arms race.

To test these hypotheses, we investigated the effects of such introductions on the structure of modern communities by answering three overarching questions: (1) What are the odds that pathogens from the past succeed in establishing themselves in a modern community? (2) What would be the impact of invaders on the diversity of invaded communities? (3) What are the features affecting the success of a time-travelling invasion?

## Results

### Probability of a successful invasion event

We quantified the success of an invasion (from the invader's perspective) in terms of how long the invader and its descendants (i.e., lineages originating from the invader's population since time of invasion) persisted in the communities. For each replicate, we therefore compared the persistence of pathogen lineages in a control community to that of an invader lineage in an invasion community, starting from time of invasion. For brevity, we henceforth refer to all the pathogens in the invaded communities other than the invader itself as 'native' pathogens. Each control-recipient community pair followed the same co-evolutionary path until the invasion event. Thus, the recipient and respective control communities at the pre-invasion stage were identical, thereby avoiding potential biases arising from stochastic events occurring before the invasion (Fig 1; see also online *Methods*).

To correct for potential biases derived from variation in the invader's initial population size (number of individual pathogens 'injected' into the modern community), we limited the comparison to simulations where there was at least one native pathogen with a population size $N$ at least as large as the invader's, and we compared persistence only between the invader and native pathogens when $N_{native} \geq N_{invader}$. On average, the invader was more persistent than 33.6% (± 39.1% standard deviation) of native pathogens. In 27.7% of simulations, the persistence of the invader's lineages was > 50% of the pathogen's in the control, while it was > 90% of the native pathogen's in 21.8% of simulations (Fig 2A).

These results suggest that an invader commonly becomes successfully established in the recipient community. Also, 3.1% of invaders were exceptionally successful, with their lineages persisting for at least 50,000 'updates' (that is, time units in *Avida*) after initial invasion (Fig 2B–2D; we used 50,000 updates after the invasion to compare between replicates because this is the minimum simulated time after an invasion). For comparison, the average (post-invasion) persistence of all other ('native') pathogen lineages from the invaded community across all simulations was 10.0 (± 20.2) updates, and only 5.4% of all native pathogen lineages survived > 50,000 updates after the invasion. Considering that invaders account for only

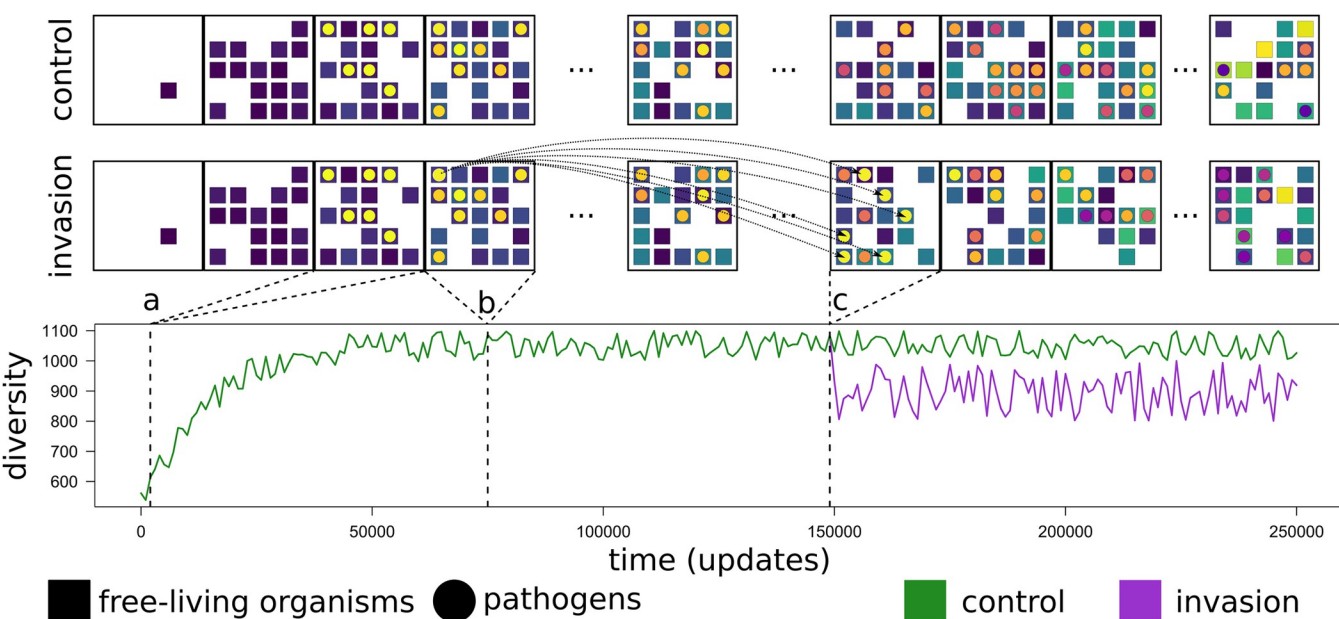

**Fig 1. Schematic representation of the simulation framework.** The scheme refers to a single pair of control *versus* invasion simulations. The two sets of contiguous squares at the top of the panel represent different, subsequent snapshots of digital communities. Free-living organisms (digital hosts) are represented as small, coloured squares, while pathogens are represented as coloured circles. Different colours correspond to different species emerging throughout the evolutionary progression of the simulations. Each pair started with the same seed from a single ancestral digital host (**a**), with a small population of digital pathogens being injected in both communities at the same moment and in the same locations. Therefore, the control and invasion communities evolved in the same way until the moment when we injected a population of time-travelling pathogens sampled from one of the community snapshots (**b**) into one future community (**c**) of the invasion time series. From that moment onward, the communities started to diverge (illustrated in the lower panel, with diversity change as an example metric). We let the two simulations run for 250,000 updates, and then we explored the potential effect of time-travelling invaders on the invaded communities by comparing the different trajectories of diversity and complexity (green *versus* purple line in the lower panel).

0.03% of all pathogens at the time of invasion, the chance of observing so many highly persistent pathogens if they were ecologically equivalent to the others is near zero (exact $p = 1.2 \times 10^{-168}$). In other words, there is a high likelihood that an invader's lineage becomes persistent over evolutionary time scales.

We obtained almost identical values for overall pathogen persistence (from invasion to the end of simulations) in the control communities, with an average persistence of pathogen lineages = 10.1 ± 20.3 updates, and only 5.5% of pathogen lineages persisting for at least 50,000 updates. These results demonstrate that although invaders can be exceptionally persistent, they normally do not reduce the overall persistence of native pathogens. In such simulations, the invader's lineage tended to become predominant in the pathogen community, accounting for 53.3% (± 41.1%) of extant pathogen species and individuals on average at 50,000 updates (Fig 3).

### Effects of time-travelling invaders on communities

We explored the ecological effects of time-travelling invaders by comparing the diversity (individual genotype richness and Shannon's *H* genotype diversity) and abundance of free-living distinct genotypes (e.g., Avidian 'species') in recipient communities compared to their control counterparts following invasion. For both diversity and abundance, we computed the area under the curve of the target measure *versus* time for both control and invasion experiments, and then quantified the relative change as the percentage difference between the two areas. For simplicity, we henceforth refer to diversity 'loss' and 'gain' in recipient communities as the

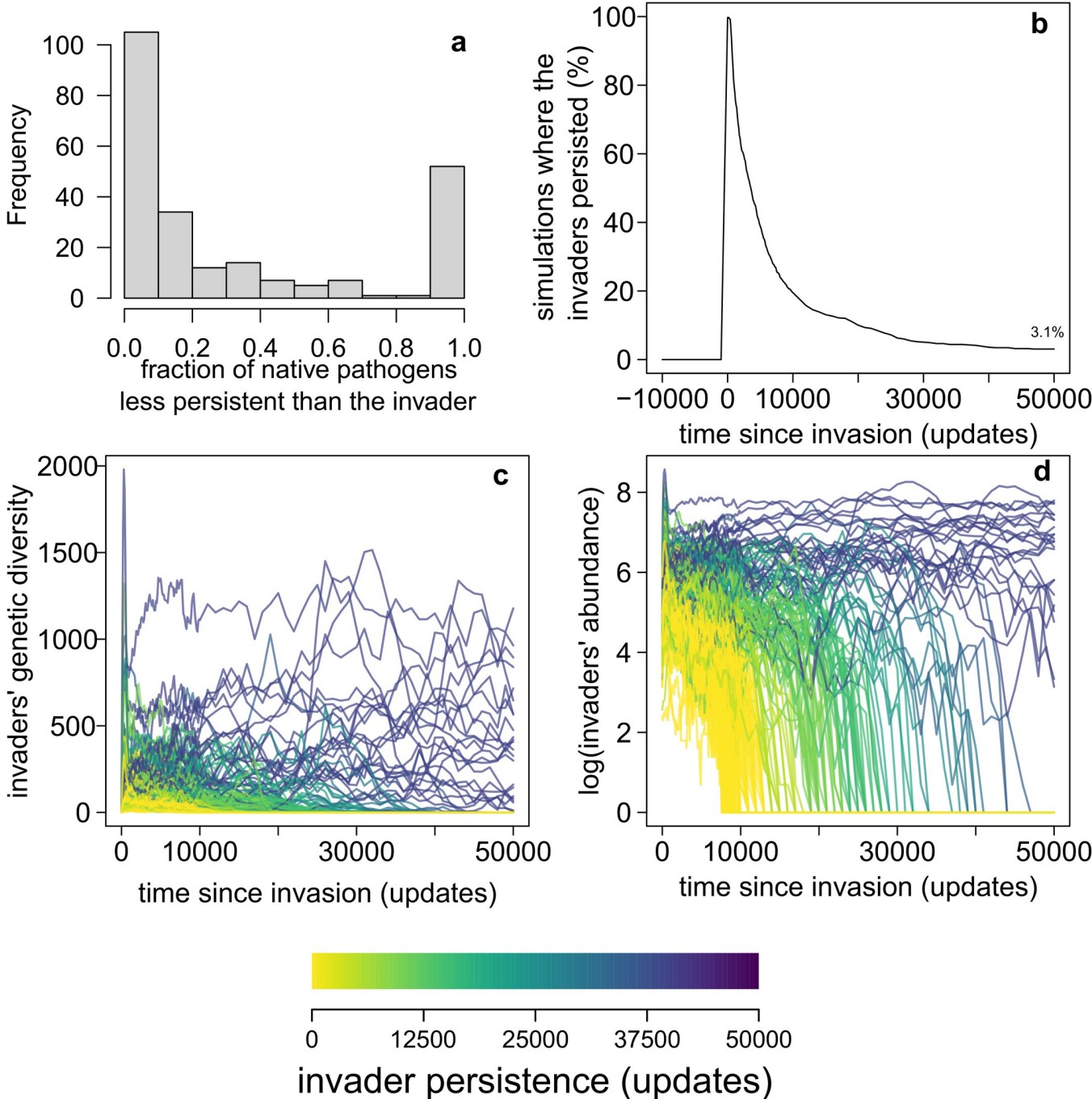

**Fig 2. Persistence of time-travelling invaders in recipient communities.** (**a**) Invader persistence compared to that of native pathogens, expressed as the distribution of the fraction of native pathogens less-persistent than the target invader across all simulations. 'Persistence' is the number of updates in the simulation where a target pathogen's lineage persisted from the moment of invasion to the end of the simulation. In most cases, almost all native pathogens are more persistent than the invader. However, in a non-negligible fraction of simulations, the invaders were more persistent than most of the local pathogens. The bimodal histogram can be explained by the phenomenon of when invaders manage to establish and persist in the simulations, they also tend to become predominant in the pathogen community, possibly leading to the extinction of most pre-existing lineages. Thus, the most common situations were those where either invaders went extinct rapidly (being less-persistent than all other pathogens), or where they instead took over the pathogen community (being more persistent than all other species); invader persistence was strongly linked to their ability to outcompete and drive native species to extinction. (**b**) Only a few invasions resulted in a prolonged persistence of invader lineages (only 3.1% of experiments at the end of the simulations). (**c,d**) Genetic diversity (number of extant genotypes at a given moment) and abundance (number of living individuals) of invaders in all the simulations. Colours represent the different relative persistence of invaders, i.e., the total number of updates the target invader's lineage survived in the simulation.

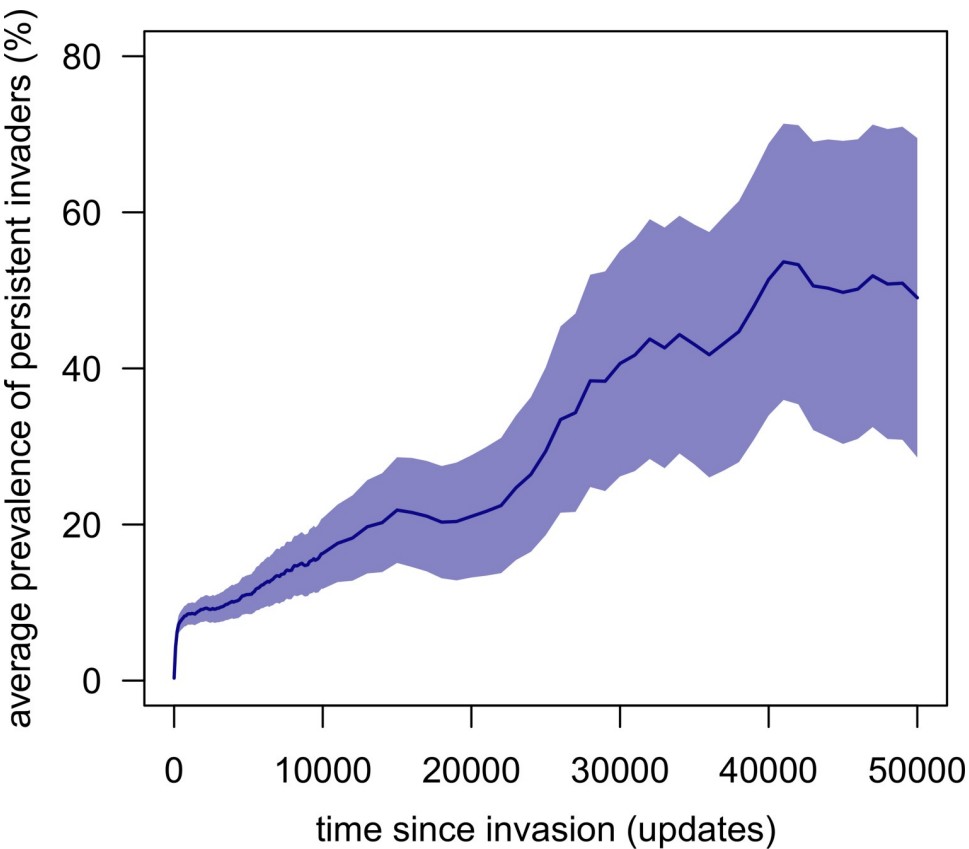

**Fig 3. Prevalence of extant invaders in the invaded communities across all simulations.** *Prevalence* is the percentage of invader species (invader and all its lineages) in the recipient pathogen community compared to all pathogen species. Solid lines are the mean values of the simulations, while shaded areas are 99% confidence intervals. Means and confidence intervals is computed every 1000 updates and includes only the invaders that persisted at least through the target update.

difference in diversity observed in the invasion compared to control simulations. Over a macro-evolutionary time scale, the system will readjust, with diversity affected by available resources, and hence the differences between the invaded and the control communities are ultimately buffered. Therefore, we focused instead on the ecological time scale and compared diversity in the first 2500 updates after invasion (corresponding to 1% of total simulation time). We report the results for species (i.e., genotype) richness only given that, within each community, this was highly correlated with individual abundance, phylogenetic diversity (computed as Faith's phylogenetic diversity), and evenness (S1 Fig; $R^2$ of relative loss of species richness *versus* loss of abundance, phylogenetic diversity, and evenness: 0.58, 0.99, and 0.70, respectively). We also explored whether the choice of the ecological time frame affected the results by replicating our analyses for different post-invasion windows with sizes spanning 500 to 5000 updates. Because the results were robust to this choice (S2 Fig), we report only the results for the 2500-update window.

Overall, invaders had a negligible effect on the diversity of free-living organisms. Invasion simulations resulted in less-diverse free-living communities than their respective controls in 54.8% of cases, but the relative difference in diversity between invasion and control simulations was slim (0.3% ± 2.9% on average, based on comparison of area under the curves of diversity *versus* time/updates; Fig 4, yellow line). However, focusing selectively on simulations

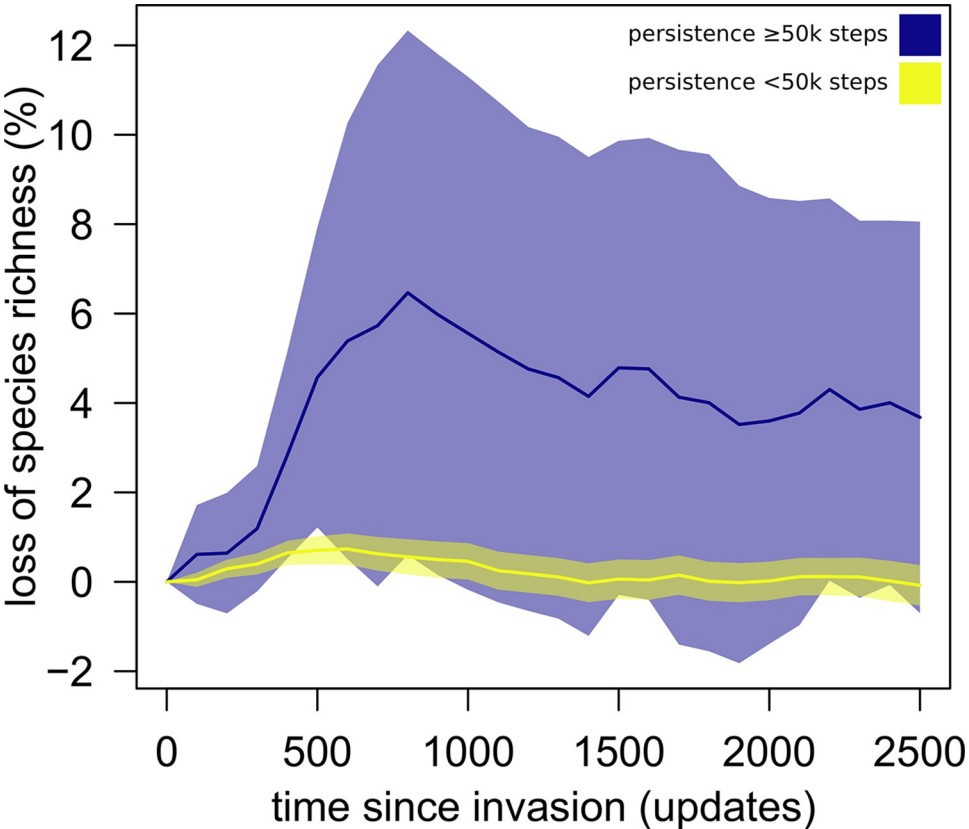

**Fig 4. Relative loss of species richness (invasions *versus* controls).** Solid lines are the mean values of the simulations with high and low persistence, while shaded areas are 99% confidence intervals. The values represent the pairwise % change in richness calculated as $100 \times (div\_con - div\_inv)/div\_con$ for each timepoint, where $div\_con$ and $div\_inv$ are species richness recorded over time in the control and the invasion treatments, respectively.

where invader lineages persisted for at least 50,000 updates, we observed a detrimental effect of invaders on diversity in 89.5% of cases, with relative losses as high as 31.9% compared to controls (4.1% ± 7.3% on average; Fig 4, blue line), and a maximum recorded gain of only 1.0%. By contrast, across all invasions, we recorded events of diversity gain up to 12.0%, possibly resulting from an escalating co-evolutionary arms race of hosts and pathogens [33,34].

## Diversity changes in time-travelling invasion simulations *versus* background expectation

Differences in diversity trajectories between control and invasion simulations might not necessarily imply a direct effect of invading pathogens on control communities. In fact, we cannot exclude *a priori* that running independent, alternative replicates of a community's future from a given point in time would produce comparable differences even without selective manipulation (e.g., the injection of time-travelling invaders). To rule out this potential problem, we ran an additional set of paired simulations where we did not translocate the invader at time of invasion. That is, instead of having paired control-invasion simulations, we ran paired control-control simulations. For these, we applied the same settings of the main experiment, and then randomly reduced the set of simulations to match the number of replicated pairs in the main simulations (*n* = 748, after removing simulations where pathogens went extinct before the invasion; see Methods). By contrasting two alternative futures of the same control

simulations, we could quantify the expected range of variation in free-living species diversity in the absence of external disturbances (i.e., time-travelling invasions). The direction of diversity change in the control-control simulations is necessarily arbitrary. In our experiments, we identified the reference simulation in each pair *a priori* (i.e., before each run).

The control-control experiment provided strong support for the primary role of invasions as drivers of diversity change. In fact, the range of variation in average diversity change between control-invasion simulations was 2.5 times greater than that observed in control-control simulations, with relative change in the latter varying from 11.4% loss to 6.11% gain (S3 Fig).

In 31.6% of the highly persistent invasions, the relative diversity loss was greater than the upper 95[th] percentile of the relative diversity change observed in the control-control simulations, indicating strong statistical support for the hypothesis that the diversity changes observed in the invasion simulations cannot be explained by the expected stochastic variation in replicate community dynamics.

## Diversity changes in time-travelling invasion simulations *versus* contemporary pathogen enrichment

Another potential confounding effect might arise from the increase in pathogen density during the re-injection of ancient pathogens into 5% of grid cells of the modern communities. That is, we need to address to what extent the observed ecological changes are due simply to the sudden increase in pathogen density following the time-travelling invasion *versus* the ancient origins of the invaders themselves. We therefore ran another experiment in which we contrasted control simulations (with no invasions), with simulations where a contemporary pathogen randomly selected among the extant ones was reinjected into 5% of grid cells (at a random time between 150,000 and 200,000 updates). For these contemporary invasion simulations, we used the same model setup and number of replicates as in the time-travelling invasion experiment.

The exercise confirmed the fundamental importance of the evolutionary gap between ancient invaders and the modern communities in determining ecological change. In fact, the range of variation in average diversity change between control/time-travelling invasion simulations was 1.9 times greater than that observed in control/contemporary invasion simulations, with relative change in the latter varying from 13.72% loss to 9.34% gain (S3 Fig).

## Explaining invasion success and its effects on invaded communities

We explored the pairwise relationship between various (pre-invasion) features of both the recipient community and the invader, and the outcomes of the invasions in terms of invader persistence, diversity loss, and absolute diversity change (i.e., absolute percentage change in species richness compared to control, including both diversity loss/gain). In particular, we considered the potential effect on invader persistence and invasion-induced diversity change of: (*i*) the size of the simulated world; (*ii*) the net amount of available resources; (*iii*) the diversity (i.e., number) of distinct available resources; local, (*iv*) free-living, and (*v*) pathogen species richness, (*vi*, *vii*) abundance, and (*viii*, *ix*) density (number of individuals per cell in the simulated world) of the invaded communities at time of invasion; (*x*) number of updates the invader's lineage originally persisted pre-invasion; (*xi*) mean and (*xi*) total invader abundance across all updates in which the invader was present pre-invasion; (*xiii*) invader abundance in the source community from which it was 'sampled' for future invasion; (*xiv*) number of tasks done by the invader, and (*xv*) maximum task complexity; (*xvi*) the invader's original generalism (number of different host species infected by the invader in the source community); (*xvii*)

the median invader's generalism in the 2500 updates following the invasion; (*xviii*) mean phylogenetic distance between invader and pathogens in the recipient community; (*xix*) time difference between the moment the invader was sourced and invasion time; (*xx*) invader's evolutionary age (number of updates from beginning of simulation) when sampled from the source community; and (*xxi*) recipient community's evolutionary age. We provide justification for the focus on each specific variable in Methods (in the section 'Hypotheses of the determinants of invader success'). We quantified relationships using Spearman's rank correlation coefficient $r_s$ (S1 Table).

We found no direct associations between the 'environmental' features of the simulated worlds (i.e., world size, abundance, diversity of resources) and the outcomes of the simulations. Conversely, we identified various positive associations between the invaders' features and their persistence in the recipient community (Fig 5). In particular, invaders capable of doing many complex tasks tended to persist longer ($r_s = 0.56$ for both task complexity and diversity *versus* invader persistence). Successful invaders in their 'first' pre-invasion existence had a higher probability of enduring re-emergence in recipient communities ($r_s = 0.48$, 0.47, 0.45 and 0.39, respectively, for mean and total pre-invasion abundance of invaders, abundance in the source community, and pre-invasion persistence). Consistent with the relationship observed for task complexity, the evolutionary age of invaders correlated positively with persistence ($r_s = 0.29$). Invaders that were generalist in the source (ancient) communities (i.e., capable of using many different hosts) also tended to be more persistent ($r_s = 0.24$). This is consistent with the relationship observed for task diversity, which contributes to generalism due to the task-matching criteria determining pathogen infection in *Avida*. In turn, highly persistent invaders tended to use a larger variety of hosts after their reintroduction into future communities ($r_s = 0.70$). Conversely, pathogen richness, abundance, and density in the invaded community contributed negatively to invader persistence ($r_s = -0.14$, -0.24, and -0.36, respectively). This might be explained by the potential competition between invaders and native pathogens, which in turn suggests that a community rich in pathogens might be less at risk of invasion.

The density of free-living species also correlated negatively with invader persistence ($r_s = -0.25$), which supports evidence for the dilution effect of high host diversity on the prevalence and spread of pathogens [35,36]. In accordance with the positive effect of invader age, persistence decreased with increasing phylogenetic distance between the invader and the native pathogen community ($r_s = -0.22$), meaning that ancient pathogens unrelated to modern ones would have a low probability of becoming successful. Similarly, invaders from source communities distant in time from the invaded ones had a lower probability of persisting than did invaders from more contemporary communities ($r_s = -0.28$).

Counter to our expectation (see section 'Hypotheses of the determinants of invader success' in the Methods), none of the considered variables correlated with (signed) diversity change ($r_s$ varying between -0.14 and 0.09). We found slightly stronger correlations when focusing on absolute diversity change; free-living species and pathogen richness, abundance, and density correlated negatively with absolute change, consistent with the patterns observed for invader persistence—$r_s = -0.16$, -0.16, and -0.22 for richness, abundance, and density (free-living species), and 0.33, 0.34, 0.34 (pathogens), respectively.

Invader persistence also correlated weakly with both signed and absolute diversity change ($r_s = -0.16$ and 0.19, respectively). Thus, persistence increased the risk of diversity loss and change in general, but we could not identify a clear relationship between how long an invader persisted and the resulting impact on the diversity of the invaded community (S4 Fig). However, we found substantial differences in the persistence of invaders in simulations leading to either increases or decreases in diversity > 5% compared to the control ($n = 23$ for losses;

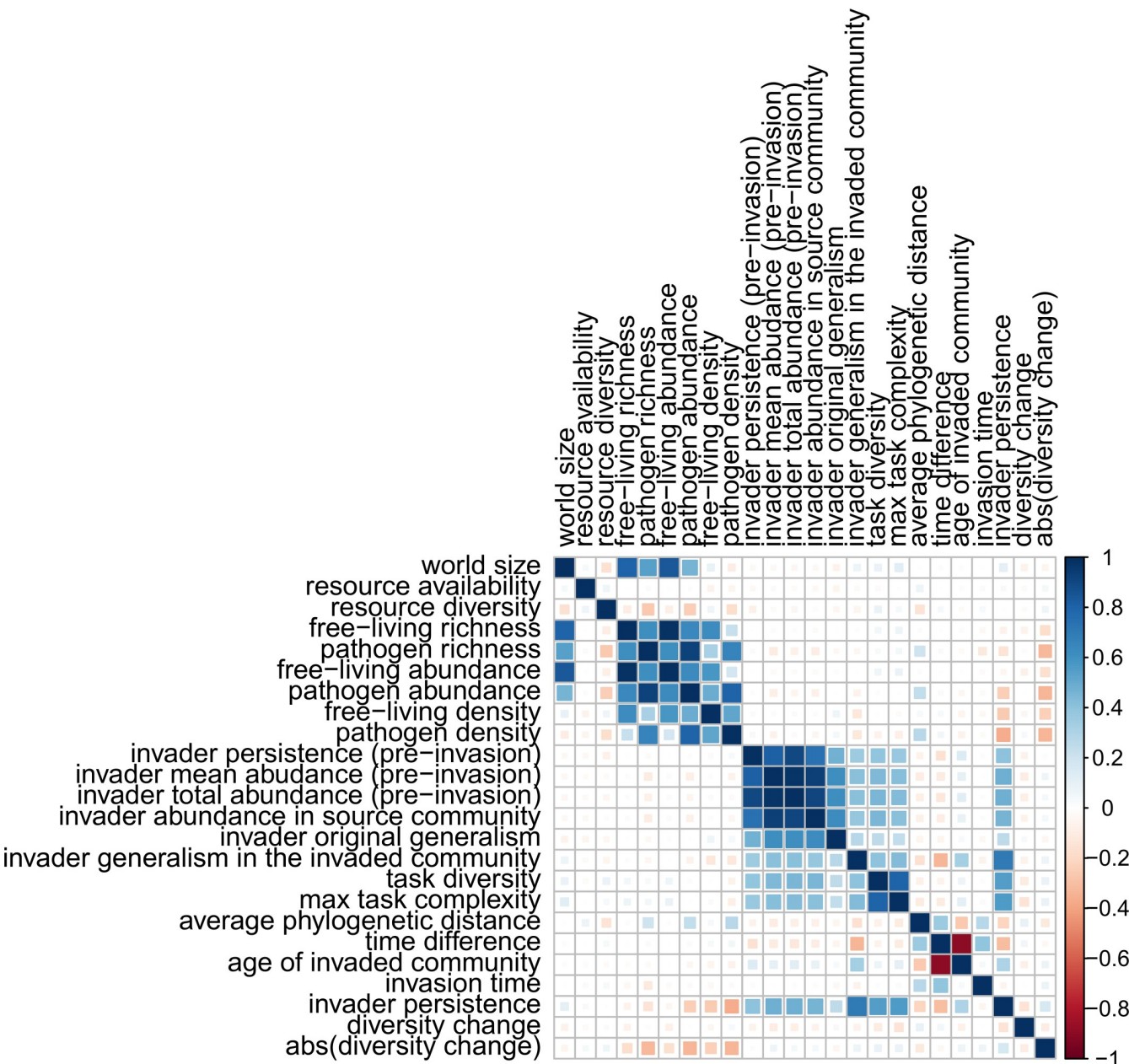

**Fig 5. Pairwise correlations (Spearman's rank correlation coefficient $r_s$) between features of the invader and of the recipient community, and the outcomes of the invasion.** Invasion outcomes are measured in terms of persistence of invaders and their descendants in the invaded communities, and relative changes in diversity in the invasion simulations compared to control simulations.

$n$ = 13 for gains). While diversity loss occurred during moderately to highly persistent invasions, gains occurred exclusively when invaders failed to persist (Wilcoxon $p = 3.3 \times 10^{-5}$; Fig 6). Together, this means that while the successful establishment of a time-travelling invader does not necessarily reduce diversity, even ephemeral invasions can precipitate substantial biodiversity loss, and persistent invaders are unlikely to have any benefit for local biodiversity.

Analogous treatment of the other variables potentially affecting invader persistence revealed patterns consistent with those above in the context of the pairwise comparisons (summarized in Fig 5). Specifically, all the measures of invader pre-invasion success, as well as invader task

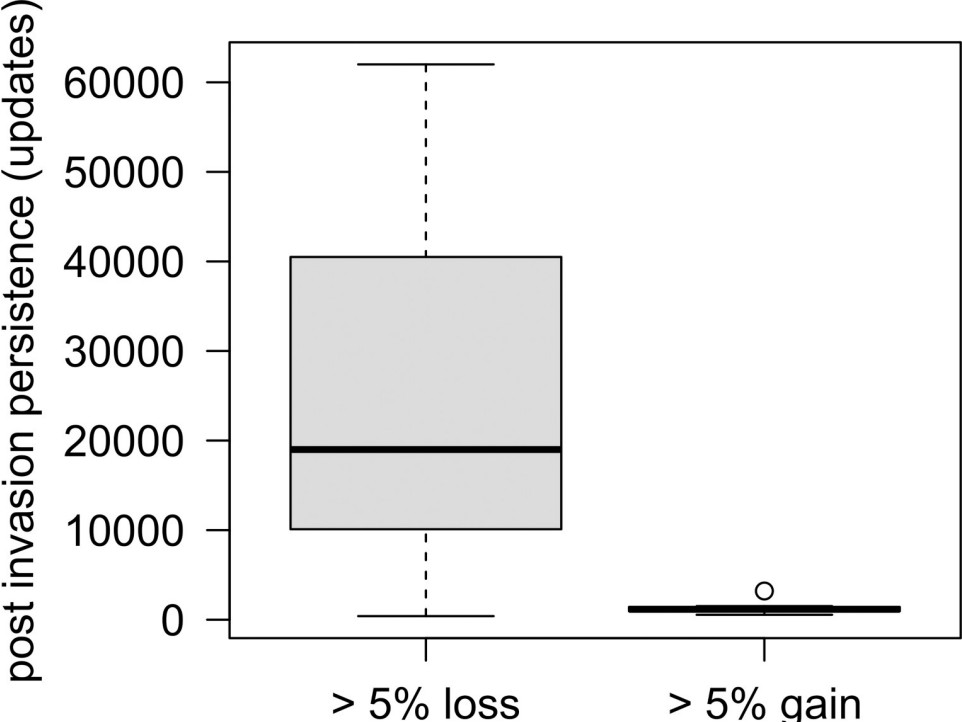

**Fig 6. Comparison of invader persistence in simulations where we observed either losses or gains in free-living species diversity (compared to control simulations).** Type I probability derived from Wilcoxon double-sided test ($n_{\text{high loss}} = 23$; $n_{\text{high gain}} = 13$), $p = 3.3 \times 10^{-5}$. Boxes indicate first and third quartiles, horizontal lines indicate median values, whiskers indicate largest/lowest points inside the range defined by the $1^{\text{st}}$ or $3^{\text{rd}}$ quartile + 1.5 times the interquartile range and circles indicate outliers.

complexity and diversity, were higher in simulations leading to losses than in those leading to gains (S5 Fig). Additionally, we found that in simulations leading to losses, free-living species richness and abundance in the recipient communities at the time of the invasion were substantially higher compared to simulations leading to gains. This suggests that high-diversity communities might be more at risk of being affected negatively by time-travelling invaders than low-diversity communities. Finally, although parasite diversity, abundance, and density in the invaded community tended to reduce invader persistence, we could not statistically differentiate these parameters in invasions leading to diversity loss from those leading to diversity gains. The latter result might indicate that while native pathogen communities can be fundamental in determining whether invaders will establish and persist in invaded communities, they might play a lesser role in modulating the invaders' impacts on the free-living community.

## Predicting invader persistence and potential impacts on modern communities

We devised a model to predict the potential persistence of an invader in a modern community, as well as its possible effects on the diversity of the invaded community. For this, we tested different multivariate approaches (e.g., generalized linear mixed models, structural equation modelling, machine learning) using both the invaders' (post-invasion) persistence, as well as the signed and absolute diversity change in control *versus* invasion simulations as the dependent variables—we also tested the 17 features of the invader and the invaded community described in the previous section as independent variables. In the models for diversity change,

we added invader persistence in the invaded communities to the list of independent variables. We obtained the top-ranked models using random forest regression [37], but even these explained only a modest amount of variance in the dependent variables (19.2%, 12.4%, and 18.0% for invader persistence, signed diversity change, and absolute diversity change, respectively). Among the variables we considered, parasite diversity, density, and abundance were among the most important predictors in all three models (S6 Fig).

## Discussion

Speculation has so far dominated the discussion of the potential threats ancient re-emerging organisms will have on modern communities [16–21,38]. For the first time, we have taken advantage of the flexibility and realism of *in silico* evolution to provide an extensive exploration of the ecological risk of time-travelling pathogens. We show that such invaders might in fact commonly establish in modern communities and sometimes promote non-negligible ecological change. In the worst, but still plausible case, the successful invasion of a single ancient pathogen reduced the richness of recipient communities by 30% relative to a non-invaded control.

Although there is a low risk of pathogen re-emergence, the inferred probability and frequency of their catastrophic repercussions are in fact much higher than our simulations might suggest. Focussing exclusively on the average effect of invasion attempts is deceptive, even if most invasions are ultimately unsuccessful. What makes a difference to real-world ecosystems is not many near misses, but the potential of even low-probability 'black swans' occurring that can individually produce broad-scale, irreversible changes [39]. Risk does not emerge solely from the chance of the event occurring, but from the combination of probability and the magnitude of the event's potential effects [40]. From that perspective, our results are worrisome, as they point to an actual risk deriving from the rare events where time-travelling invasions produce severe ecological impacts.

Furthermore, the few successful and/or impactful invasions observed in our simulations might be deceiving. We simulated a single invasion attempt by a single pathogen genotype. That is, in our experiments, the probability of incurring an effect was related to a single invasion attempt. But, in fact, long-term cryogenic formations in ice sheets and glaciers can host 10–100 bacteria ml$^{-1}$, and there can be up to $10^7$ cells g$^{-1}$ in frozen ground and buried soils— today, there are an estimated $4\times10^{21}$ cells eluted by ice melt from non-Antarctic glacial systems annually [41,42]. Glacial environments also host a density of virus-like-particles in the order of millions to tens of millions per millilitre [43]. Thus, even if only 0.1% of attempts precipitate substantial ecological change in communities, the sheer volume of exposed cells suggests that a non-negligible ecological change boosted by ancient invaders could be happening already. Additionally, the risk of invaders to be detrimental to modern communities in our simulations increased with the ability of invaders to establish themselves for a long time post-invasion, which, in turn, was positively associated with the persistence and abundance of the invader in its 'first' existence. This makes the real-world potential ecological risk substantial, because such pathogens with a successful past history are also more likely to show up in the 'frozen record'.

Although *Avida* has proven to be capable of remarkable realism in simulating evolutionary and ecological processes [28,29,44,45], it is clear that there are fundamental differences between Avidian digital worlds and our own. One important aspect is that simulation updates and generations in *Avida* cannot be directly translated into a consistent measure of time, which might cast doubt on how our experiments align with real-world settings. In other words, how does the number of updates in the simulations compare to the estimated number of intervening generations in melting ice, for example? Although we cannot provide a

definitive answer to this question, we can use Fermi estimations incorporating mutational distances between evolved Avidians and their ancestors. We found an average of ~400 substitutions along the lineage of the most abundant parasite to the ancestor in these runs (S7 Fig). If we consider a single fixed mutation along Avidian lineages as equivalent to a single substitution in *Escherichia coli* (a conservative assumption in some respects, considering that mutations in *Avida* are much more 'functional' than single nucleotides), and use an estimated 0.0001–0.0002 mutations per generation of *E. coli* to scale our observations, we can estimate that the length of our simulations roughly correspond to ~2–4 million generations of microbial evolution. Thus, our invaders (sampled from populations in between 50,000 and 150,000 updates over a total of 250,000 updates) could come from an approximated equivalent of 400,000–2,400,000 bacterial generations. Considering that viable bacteria have been recovered from ~750,000 year-old glacial ice [46] and that natural microbial populations typically have long generation times, our simulations fall within a more-than-reasonable (co)evolutionary time frame.

While our model represents the first attempt to quantify the risks from time-travelling pathogens, it also introduces a flexible modelling framework that could be ideally used in combination with other artificial life systems and evolutionary simulation platforms. These combinations would not only be able to test the validity and generality of our findings, they could also investigate countless fundamental questions of future risk to biological communities from emerging organisms. Such questions of course extend to the study of 'spatial' biological invasions. Future work could investigate how and to what extent the mechanisms regulating time-travelling invasions differ from those dictating spatial invasions. More generally, our study offer another concrete example on how 'digital twins' can help exploring future, complex scenarios and test hypotheses for which we have no empirical data [47–49].

Our findings corroborate the general consensus that biological invasions are almost impossible to predict, due to the complexity of the ecological and evolutionary factors involved [8]. Specifically, our analyses revealed that multiple variables related to features of either the invaders or the invaded communities might have complex and contrasting effects on the potential outcomes of an invasion. Such complexity prohibits devising a model capable of predicting the outcome of an invasion in general terms. While we did attempt to apply various multivariate approaches to explain diversity loss and invader persistence as a function of variation in all features compared in pairwise simulation outcomes, these showed only modest predictive power. Considering that *in silico* communities are ideal for this purpose because they pose no limits in the number of variables and amount of data one could retrieve and include in a model, our struggle suggests that accurately predicting the potential impacts of time-travelling invasions in real-world settings might be exceedingly challenging.

Nonetheless, our results provide insights useful for improving preparedness by identifying vulnerabilities. In particular, they support empirical observations that diverse and dense communities are less susceptible to invasions than depauperate ones. Indeed, the outcome that high pathogen diversity and abundance can decrease the probability that time-travelling invaders establish in modern communities emphasises the importance of preserving not only free-living species, but their pathogens and parasites as well [50,51].

While our work focuses on the ecological implications of time-travelling invasions at the scale of the microbial community, ancient pathogens are also a potential threat to human health, both as direct agents of disease and as potential sources of novel zoonotic risks, a highly relevant topic in light of the recent COVID-19 pandemic [52,53]. The occurrence of similar events is intrinsically unpredictable, but modelling frameworks such as ours can offer a unique opportunity to improve risk assessment by formally quantifying the odds for ancient invaders to thrive in modern communities.

## Online methods

### Simulation setup

Using *Avida* (version 2.14; avida.devosoft.org), we generated 100 different worlds with random size between 2500 and 15,000 grid cells, which we held constant throughout the experiment. The configurations also covered a broad range of different (stochastic) environmental settings, spanning environments with a few scarce resources, to settings with many abundant ones. We created nine different resource pools and randomly assigned each of them to the standard nine logical operations that the *Avida* digital organisms can do [45]. The same resources were also assigned randomly as by-products to logic functions, with an input-output ratio randomly selected between 0 and 0.5. This effectively creates a random biochemical network unique to each configuration. To create further environmental heterogeneity, we set the input and output of each resource randomly every update.

We then ran two sets of 1100 simulations (11 replicates per *Avida* configuration). In the first set, we let communities evolve from a single, free-living ancestor and a single pathogen ancestor over 250,000 updates. We recorded the complete eco-evolutionary history of each community by taking snapshots of organism diversity (richness) and abundance, as well as of host-pathogen interactions. We then randomly selected a community snapshot between updates 50,000 and 150,000, and from that community we randomly stored one pathogen genotype (with selection probability proportional to the pathogen's abundance in the community). In the second set of simulations, we replayed the exact eco-evolutionary history for each community as in the first set up until a randomly selected step between updates 150,000 and 200,000, where we then injected a small population of the previously stored pathogen from the past into 5% of the *Avida* world's cells (each pair of communities from the two sets of experiments were identical at each step up until the time-travelling invader was introduced). We let the simulations complete (until update 250,000), and then compared their outcomes with those of the respective control experiments to assess the fate and potential impacts of time-travelling invaders on the recipient communities.

In both control and invasion simulations, we recorded communities every 100 updates for the first 10,000 updates after the time of invasion, and every 1000 updates thereafter. In all experiments, the lineages of the successful invaders were already extinct at the time of invasion (i.e., the selected minimum time-lag between the source and recipient communities was enough to ensure a complete phylogenetic separation between the invader and the recipient community—no species in the modern community could be sourced from the invader or its descendants). To ensure a fair comparison between simulations, we excluded all the simulations from the final analyses where pathogens went extinct before the invasion, leaving 748 simulations.

### Hypotheses of the determinants of invader success

The potential threats ancient pathogens pose to modern ecological communities are mostly unresolved. The closest empirical examples available are rare and isolated episodes of re-emergence of recent, virtually eradicated diseases [54]. Similar events of reappearance of bygone pathogens often result in severe infections, such as when several people in Siberia were exposed to anthrax spores likely coming from carcasses of infected, long-dead animals buried in permafrost, which were released by ice thawing [55]. There could potentially be many unrecorded cases where pathogens severely affected host communities (particularly for bacterial communities analogous to our model agents), as well as undocumented examples where contact between a bygone pathogen and a modern ecological community resulted in either the

death of the pathogen, or in its benign re-establishment. However, this knowledge gap does not prevent the opportunity of formulating and testing *in silico* hypotheses regarding the processes leading to a higher probability of successful establishment in, and the noxious consequences for, recipient communities based on recorded biological invasions.

While opportunities for biological invasions have grown exponentially with globalization [56], most invasion attempts are expected to fail [57,58]. For instance, while ballast water is a main vector of marine invaders, the quantity and diversity of organisms they carry are incomparably larger than the number of successful establishments [59,60]. We therefore hypothesize the same pattern for time time-travelling pathogens—most invasions should either fail early, or settle without appreciable effects on the recipient community.

One of the main determinants of success of a biological invasion is environmental filtering —community sorting along abiotic gradients [61]. However, the role of environmental filtering might be less important for time-travelling invaders because we expect relatively less temporal variation than spatial variation in environmental conditions, depending on the temporal scale of investigation. This is especially true in our simulations because we did not simulate changing environmental conditions through time. This might lead to a higher probability of successful establishment for time-travelling invaders in our simulations compared to spatial translocations of organisms between different simulated worlds.

Organisms in *Avida* interact with the environment not just by consuming resources, but also by transforming them and releasing metabolic by-products. Consequently, even if a given simulated environment receives a constant input of abiotic resources, the environmental conditions experienced by the invading pathogen at the time of invasion will not necessarily be identical to those it experienced in the source community. This arises because the overall diversity and availability of resources are determined by a combination of steady inflow and outflow of resources, and of the community in place at any given time (which uses and transforms the available resources, generating variable ecological complexity and, to some extent, trophic structure) [62]. This might therefore facilitate environmental filtering when organisms move across time within an otherwise identical environment.

Various attempts to identify predictors of success in biological invaders have shown them to be largely contextual, thereby precluding generalizations [63,64]. However, our approach offers at least three major advantages over existing studies. The first is providing exhaustive ecological and evolutionary information about both the invader and the invaded community. Second, our ability to simulate two parallel futures (with and without invaders) from the same starting point provides a standard quantification of invader success. Third, by replicating the same kind of organisms and consistent (albeit, varying) environmental settings, there are fewer potentially confounding factors stemming from taxon- and site-specific features. All considered, we expect to be able to identify at least some general determinants of invader success (which we quantified as persistence of the invader and its descendants in the invaded community). We list below a set of potential hypotheses, with the Roman numerals in parentheses referring to the actual parameters used in the analyses described in section "Predicting invaders' persistence and their potential effects on the invaded communities".

We did not have any specific expectation on the *direct* effects of the environmental variables (world size, abundance, and diversity of resources; *i–iii*) on the persistence and ecological impacts of invaders on the invaded communities. However, we expected these variables to have potential indirect effects on the outcomes of the invasions by affecting the diversity and complexity of the evolved communities. We specifically examined how the richness and abundance of native species affected the chances of an invader settling and altering the invaded community (*iv–ix*). We considered two alternative hypotheses regarding the effect of diversity on invasions. The 'rich get richer' hypothesis, based on the observed correlation at a broad

spatial scale of the diversity of alien species *versus* local diversity [65], suggests that communities hosting many species and individuals are more prone to invasion. An alternative hypothesis is that rich communities might reduce the chances of invasion due to a stronger competition for resources or due to a higher chance for the existence of antagonistic mechanisms [66,67]. Since we focused on pathogens in our virtual experiments, we can allocate the two hypotheses separately to free-living host species and their pathogens. That is, we expect that a high diversity of potential free-living hosts promotes invasions, while a high diversity (and abundance/density) of pathogens reduces the chances for an invader to establish due to stronger competition for resources.

In addition to competition with native species, invading pathogens need to find an appropriate host [50]. In *Avida*, pathogens can infect a host if they do at least one of the same logic functions as the free-living host organism. The arms race between hosts and parasites in *Avida* also leads to an increasing complexity, because doing novel, complex tasks permits hosts to escape from previously infective pathogens [27]. The resultant hypotheses predict that invaders capable of doing many tasks (*xvi*), and possibly complex ones (*xv*), and that are hence also able to infect many host species in the source community (*xvi*), are expected to succeed more often than specialized or 'simpler' invaders.

Similarly, if the source and the invaded community have a long temporal separation, the modern hosts might be too complex for the pathogen to proliferate. We therefore hypothesize that the chances of a pathogen invading successfully should decrease with the time difference between when the invader was sourced and when the invasion took place (*xix*), and with the recipient community's evolutionary age (*xxi*). In contrast, invasion success should increase with the invader's evolutionary age (number of updates from the beginning of runs) when sampled from the source community (*xx*).

Darwin's naturalization conundrum generates two alternative hypotheses related to how the phylogenetic relatedness of invader and native species (*xviii*) affects the invader's probability of establishment. On the one hand, we expect that increasing phylogenetic relatedness between alien pathogens and pathogens of the invaded community will increase the invader's chances of infecting modern hosts. On the other, we expect that more distantly related pathogens can more completely exploit unused resources (i.e., hosts that had evaded modern pathogens but are possibly susceptible to ancient ones). Recent work suggests that at finer spatial scales (such as in our self-contained *Avida* world), the effect of competition dominates the advantages of pre-adaptation; hence, the prediction that related pathogens should be less successful is more likely [68].

We also hypothesized that invaders more abundant in the past would have a higher probability of establishing abundant populations in the invaded communities (*xi-xiii*). This prediction is based on studies showing that invader abundance in the native range predicts abundance in the invaded community [69]. In addition to the different measures of abundance, we also hypothesized that some of the features making an invader highly persistent in the past could promote its persistence in the future invaded community (*x*) (even though this aspect does not have clear correspondence with elements of spatial invasions).

Finally, we hypothesized a positive relationship between the success of an invader (in terms of long-term persistence in the invaded community) and its effects on the diversity of the invaded community, based on the assumed positive relationship between invader abundance and its impact on the invaded communities [70,71].

## Assessing invader persistence

In each invasion simulation, we quantified an invader's persistence by the number of updates post-invasion where the invader's lineage (i.e., the invader or any of its descendants) was

present. Similarly, we quantified the post-invasion persistence of the lineages stemming from all individual genotypes belonging to native pathogens. We limited the assessment to 50,000 updates after the invasion because this was the minimum possible number of post-invasion updates (see 'Simulation setup' section). We then compared the invader's persistence with that of the native pathogens. We excluded from the comparison any native pathogen species in the invaded community with a smaller population size than that of the invader (at the time of the invasion), to remove potential biases due to differences in the size of the initial populations. The comparison was done by quantifying the percentage of native species less-persistent than the invader in each simulation. We also quantified the expected probability of any pathogen to persist for at least 50,000 updates as the overall fraction of pathogens that persisted at least that long across all simulations. We then counted how many invaders persisted for 50,000 updates, and estimated the probability of observing the same number of highly persistent invaders using the Bernoulli formula.

## Assessing diversity change

To quantify the effect of invaders on the recipient communities, we compared trajectories of free-living species diversity and abundance in the invasion experiments with the corresponding trajectories in the control experiments. We computed the areas under the curve of diversity (number of species) *versus* time (from 0 to 2500 updates since invasion, with a resolution of 100 updates) for each invasion experiment and the corresponding control experiment, and then computed the relative (%) change (compared to the control simulation). We replicated the analysis alternatively using individual abundance, phylogenetic diversity, and Shannon's diversity of genotypes ($H = -\sum_i^S p_i \log(p_i)$), with $p_i$ = proportion of individuals in the population having the $i^{\text{th}}$ genotype, and $S$ = total number of species in the population). Since the measures were highly correlated, we reported only the results for species richness.

## Comparing observed change with 'natural' variation between simulations

We explored whether the observed relative changes in diversity patterns observed in control *versus* invasion simulations could be explained by the expected 'natural' stochasticity between replicated future community dynamics starting from the same initial conditions (i.e., the same community at the same time point). For this, we ran an additional 1000 pairs of simulations using the same procedure described above, with the only difference being that instead of injecting time-travelling invaders into one of the simulations, we instead changed the random seed. In other words, we contrasted control *versus* control simulations. This allowed us to quantify the expected relative deviations in diversity trajectories between each pair of simulations, which we then compared with the variation observed in control-invasion simulations, and we assessed diversity change as we did for control-invasion simulations. However, in the control-control case, the direction of change (loss or gain) is not meaningful because the choice of which simulation in each pair is the reference scenario is arbitrary. Here, we identified the reference simulation in each pair prior to running the simulations, and did all the subsequent analyses accordingly, hence computing change as the percentage difference of the area under the curve computed for the designated reference simulation and that computed for the paired (non-reference) simulation.

## Predicting invader persistence and their potential effects on the invaded communities

We tested different multivariate approaches with the aim of devising a model to predict invader persistence and relative signed and absolute diversity change (of invasion compared to

control simulations)—generalized linear mixed models, structural equation modelling, and machine learning. All approaches produced models with low predictive power, but we obtained the best models using random forest regression (we report only the random forest results in the main text). As independent variables in all models, we used the 17 features of the invader and the invaded community listed in section "Explaining invasion success and its effects on invaded communities": (*i*) free-living and (*ii*) pathogen species richness, (*iii*, *iv*) abundance, and (*v*, *vi*) density (number of individuals cell$^{-1}$ in the simulated world) of the invaded communities at time of invasion; (*vii*) number of updates where the invader's lineage existed pre-invasion; (*viii*) mean and (*ix*) total invader abundance across all updates in which the invader was present pre-invasion; (*x*) invader abundance in the source community from which it was 'sampled' for future invasion; (*xi*) number of tasks done by the invader, and (*xii*) maximum task complexity; (*xiii*) the invader's original generalism (number of different host species infected by the invader in the source community); (*xiv*) mean phylogenetic distance between invader and pathogen in the recipient community; (*xv*) time difference between moment invader was sourced and invasion time; (*xvi*) invader's evolutionary age (number of updates from beginning of simulation) when sampled from the source community; and (*xvii*) recipient community's evolutionary age. In the models aimed at predicting relative diversity change (either signed or absolute), we also included invader (post-invasion) persistence in the set of independent variables. We developed the random forest regressions in R using the randomForest package [72], which implements Breiman's random forest algorithm [37]. For each model, we grew 1000 trees, and evaluated performance in terms of explained variance.

## Supporting information

**S1 Table. Pairwise correlations (Spearman's rank correlation coefficient $r_s$) between features of the invader and of the recipient community, and the outcomes of the invasion.** Invasion outcomes are measured in terms of persistence of invaders and their descendants in the invaded communities, and relative changes in diversity in the invasion simulations compared to control simulations.
(DOCX)

**S1 Fig. Comparing different measures of diversity change in invaded *versus* control communities.**
(PDF)

**S2 Fig. Effect of the selected ecological time scale on the quantification of invader impacts on the invaded communities.** (a) Relative post-invasion % change in free-living diversity in control *versus* invasion simulations computed in a time window of 2500 updates (*x* axis), compared to the corresponding values obtained using alternative time windows spanning 500 to 5000 updates (*y* axis, different colours). (b) Boxplots summarizing the relative post-invasion % change in free-living diversity in control *versus* invasion simulations computed using alternative time-windows spanning 500 to 5000 updates (the first, white boxplot refers to the actual time window we used in the analyses, i.e., 2500 updates, corresponding to 1% of the total number of updates in a simulation). Boxes indicate 1$^{st}$ and 3$^{rd}$ quartiles, horizontal lines indicate median values, whiskers indicate largest/lowest points inside the range defined by the 1$^{st}$ or 3$^{rd}$ quartile + 1.5 times the interquartile range and circles indicate outliers.
(PDF)

**S3 Fig. Range of diversity change observed in control *versus* control simulations (left violin plot); control simulations *versus* contemporary invasion simulations (middle violin plot); control simulations *versus* time travelling invasion simulations (right violin plot).** Boxes

indicate the $1^{st}$ and $3^{rd}$ quartiles, white dots indicate median values and whiskers indicate largest/lowest points inside the range defined by the $1^{st}$ or $3^{rd}$ quartile + 1.5 times the interquartile range.
(PDF)

**S4 Fig. Signed (a) and absolute (b) percentage change in species richness in invasion simulations compared to control simulations for different values of invader persistence (expressed as the natural logarithm of the number of updates in a simulation).** Boxes indicate $1^{st}$ and $3^{rd}$ quartiles, horizontal lines indicate median values, whiskers indicate largest/lowest points inside the range defined by the $1^{st}$ or $3^{rd}$ quartile + 1.5 times the interquartile range and circles indicate outliers.
(PDF)

**S5 Fig. Different features of invasions leading to either diversity loss or gain (compared to control simulations).** Type I probabilities ($p$) derived from Wilcoxon double-sided test ($n_{\text{high loss}}$ = 23; $n_{\text{high gain}}$ = 13). Boxes indicate the $1^{st}$ and $3^{rd}$ quartiles, horizontal lines indicate median values, whiskers indicate largest/lowest points inside the range defined by the $1^{st}$ or $3^{rd}$ quartile + 1.5 times the interquartile range. Where not otherwise specified, community properties (e.g., free-living richness, pathogen richness, etc.) refer to the recipient (i.e., invaded) community at the time of the invasion.
(PDF)

**S6 Fig. The ten most important variables in the models predicting invader persistence and their potential impact on the invaded communities.** The three models are random forest regressions aimed at predicting time-travelling invaders' post-invasion persistence (left panel), and relative signed (middle panel) and absolute (right panel) diversity change in invasion *versus* control simulations based on several features of the invaders and of the invaded communities (see Methods or main text for the full list of considered variables). Variable importance is computed as the percentage increase in the mean squared prediction errors (MSE; estimated through out-of-bag validation) following the random permutation of the target variable. Higher MSE indicate more predictive variables.
(PDF)

**S7 Fig. Distribution of the number of substitutions along the lineage of the final most abundant parasite to the ancestor in the set of control simulations ($n$ = 654).**
(PNG)

## Author Contributions

**Conceptualization:** Giovanni Strona, Corey J. A. Bradshaw, Pedro Cardoso, Nicholas J. Gotelli, Frédéric Guillaume, Federica Manca, Ville Mustonen, Luis Zaman.

**Data curation:** Giovanni Strona.

**Formal analysis:** Giovanni Strona, Luis Zaman.

**Funding acquisition:** Giovanni Strona.

**Investigation:** Giovanni Strona.

**Methodology:** Giovanni Strona, Luis Zaman.

**Supervision:** Giovanni Strona, Corey J. A. Bradshaw, Ville Mustonen.

**Validation:** Giovanni Strona.

**Visualization:** Giovanni Strona, Luis Zaman.

**Writing – original draft:** Giovanni Strona, Corey J. A. Bradshaw, Pedro Cardoso, Nicholas J. Gotelli, Frédéric Guillaume, Federica Manca, Ville Mustonen, Luis Zaman.

**Writing – review & editing:** Giovanni Strona, Corey J. A. Bradshaw, Pedro Cardoso, Nicholas J. Gotelli, Frédéric Guillaume, Federica Manca, Ville Mustonen, Luis Zaman.

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
