## [Decision Letter · Decision Letter 0]

5 Apr 2023

Dear Dr Strona,

Thank you very much for submitting your manuscript "Time-travelling pathogens and their risk to ecological communities" for consideration at PLOS Computational Biology. As with all papers reviewed by the journal, your manuscript was reviewed by members of the editorial board and by several independent reviewers. The reviewers appreciated the attention to an important topic. Based on the reviews, we are likely to accept this manuscript for publication, providing that you modify the manuscript according to the review recommendations.

Sincerely,

Rustom Antia

Academic Editor

PLOS Computational Biology

Natalia Komarova

Section Editor

PLOS Computational Biology

Reviewer's Responses to Questions

**Comments to the Authors:**

Reviewer #1: This study uses a simulation approach to investigate the dynamics of “time-travelling pathogens” (i.e., pathogens from the past, conserved in natural settings like permafrost, or artificial settings like laboratories) following their reemergence, and their impact on their potential-host community. The manuscript is very well written, and, as a non-computational biologist, I really enjoyed reading it. The study is well justified, and the methods and results are clear, except for a few points detailed below. The study is also very well conducted. The main highlights to me are that:

(1) It considers multi-hosts, multi-pathogen systems, providing insights that are difficult to get from empirical studies (lines 91-100).

(2) It includes paired control scenarios (with no time-travelling pathogen reemergence; lines 91-100).

(3) It includes an exploration of the potential mechanisms explaining the persistence of the pathogen and the effect of the pathogen on the potential-host community (lines 181-255), and all those potential mechanisms are well presented including detailed descriptions of the expected effects of each considered variable (lines 522-618).

Major comments

The way some of the key features of pathogen dynamics are modeled is either unclear or presented too late in the manuscript. Solving this issue would make the manuscript more convincing to infectious disease biologists.

(1) Host specificity: the notion of generalist/specialist is mentioned for the first time on line 207, but the relationship with task complexity is explained only on line 482.

(2) It is unclear to me how immunity is accounted for. I understand that host can evade pathogens by evolving novel tasks (if I understood well), which represent an innate barrier to infection (e.g., lack of compatibility between a host cell receptor and a virus receptor-binding domain), but I do not understand how adaptive immunity is accounted for. I am particularly thinking about potential cross-reactions between native pathogens and the time-travelling pathogen, as we expect for instance for small pox (the extinct, hopefully not time-travelling, pathogen) and monkey pox (the native pathogen, still circulating in human populations) – see Lloyd-Smith, J. O. (2013). Vacated niches, competitive release and the community ecology of pathogen eradication. Philosophical Transactions of the Royal Society B: Biological Sciences, 368(1623), 20120150. https://doi.org/10.1098/rstb.2012.0150

(3) Please clarify how “transmission” happens. When a host is infected, it sheds the pathogen on its cell grid, and other hosts can get infected if visiting this cell grid and matching with the pathogen task-wise? If that is the case, how long does a pathogen remain infectious on a cell grid?

Related to point (2) above and to the statement made lines 40-41, adaptive immunity is also (in addition to co-adaptation) a key factor in invasion by pathogens (also discussed in Lloyd-Smith, 2013).

How is a “species” (line 126) defined in the model? Is there a threshold of divergence between two individuals to consider they belong to different species?

Because the manuscript presents the results before the methods, some terms should be briefly defined earlier:

(4) Complexity (line 76)

(5) Task (line 190)

Please also include a brief definition of “Shannon’s H evenness”.

Minor comments

I suggest using “pathogen” instead of “virus” as much as possible in the manuscript as I believe the model could also apply to bacteria and other pathogens, or at least be consistent and avoid switching back and forth between the two words.

I was also a bit confused by the wording “free-living” (line 26). Could this be called “potential hosts” (e.g., at the first occurrence: “free-living (potential host) species”)?

Lines 144-… – It is a bit confusing to say “Regardless of invader persistence, invaders had a negligible effect on diversity…”, followed by “However, focusing selectively on successful invasions […], the observed relative loss of free-living species diversity increased to 89.5%”. Could the beginning of the sentence be replaced by “On average” or “overall”?

Line 328 – Add “COVID-19” as there are several pandemics happening at the moment (HIV, avian influenza…).

Line 301 – “Avida” to be italic?

Line 490 – Add “grid” in “15,000 [grid] cells” if that is correct, to make it explicit that you are not talking about organism cells.

Lines 526-528 – Is the ancient anthrax strain still circulating? If not, should not you say that this reemergence event ended up with the death of the pathogen?

Reviewer #2: This paper used simulations of the CPU-consuming artificial life forms in AVIDA to ask what happens when pathogens which went extinct are brought back into a modern age. On the whole this is a nice paper with clear explanations and figures, as well as good attempts to provide control simulations for a complicated experiment. My main criticism is that some of the caveats should be given more space in the discussion, especially the caveat that right now there isn’t a way to disentangle the confound of parasite invasion and disturbance to the community. Additionally, while simulations are critical to our understanding, the world of AVIDA is so different from our world that it is quite difficult to interpret things like % of successful invasions—we can’t know at the moment whether this has any relationship to real life. My other main criticism is that a number of important details are in the methods which should really go in the introduction, and finally that more effort should be made to situate this paper in the existing invasion simulation literature.

Specifics:

I really like the schematic in Fig 1, it is quite clear.

How does the number of intervening generations in the simulations compare to the estimated number of intervening generations in e.g. melting ice?

The colorbar label “steps” in fig 2cd is hard to interpret; the legend says “different relative persistence of invaders” which made me assume it is trying to show the x-axis data from fig 2a, but I may not be following. Please clarify.

I think Avida could be better introduced in the introduction rather than just in the methods, especially the key difference between free-living and parasitic individuals. Also, can free-living hosts also be parasitic? If so, it should be clarified how this class of organisms was dealt with.

I appreciate the goal of exploring many different simulation setups (input-out ratios, world size, etc). But this then begs the question of whether these simulation variables influenced the outcome of invasion. Perhaps these interesting variables could be added to the correlation map in Figure 5?

Why was the invasion on a random step? Doesn’t this make it harder to interpret comparisons at gen 250000, since different simulations have had a different number of generations since invasion?

“Hypothesis of the determinants of invader success” seems more like intro or discussion than methods.

One of the key hypotheses is “if the source and the invaded community have a long temporal separation, the modern hosts might be too complex for the pathogen to proliferate.” Is this possible in AVIDA? From reading the methods, a parasite only needs to be able to do the same task as a “host” to consume its CPU cycles. Testably, what % of invaders matched at least one host’s task at the moment of invasion? This would help disentangle failures due to lack of niche from failures due to intra-pathogenic competition.

I think the no-invasion control was a good choice. Another choice could have been to do a “local” (or “modern”) invasion, where 5% of the world’s cells received extant pathogens (this seems like what the “natural” variation controls was aiming at, but didn’t quite capture). The current control leaves invader presence and change in density confounded, my suggested control would remove the density confound. I don’t think this is necessary but if not done, the confounded nature of invasion and change in density should be remarked upon as a caveat in the discussion.

Put more generally, right now it is hard to pin down the effects on diversity to a cause of the specific invader, or the fact that there was generally “disturbance.”

“we limited the comparison to simulations where there was at least one native pathogen with a population size N at least as large as the invader’s,” <- this is interesting and your rationale about pop size makes sense, but it does change the scope of the experiment, because it is biasing towards worlds where we know that at least some pathogens can exist. Important caveat.

“We focused on the ecological time scale here, and hence compared diversity in the first 2500 steps after the invasion” please explain why 2500 was chosen specifically. Would the results have differed much with a 1000 or 5000 step window, for example?

I appreciate this caveat in the discussion: “Although Avida has proven to be capable of remarkable realism in simulating evolutionary and ecological processes34–37, it is clear that there are substantial differences between Avidians and real world microorganisms.” I think it could go a bit further though, because as it is, the lack of similarity means that I’m not sure we should really give much interpretation to the 3% success.

While this may be the first sim to specifically examine “time-traveling” invasions, there is a wealth of existing literature simulating invasion generally. More reference to existing work would be place this article in the existing knowledgebase.

**Have the authors made all data and (if applicable) computational code underlying the findings in their manuscript fully available?**

Reviewer #1: Yes

Reviewer #2: Yes

PLOS authors have the option to publish the peer review history of their article (what does this mean?). If published, this will include your full peer review and any attached files.

Reviewer #1: No

Reviewer #2: No

Figure Files:

Data Requirements:

Reproducibility:

References:

---

## [Decision Letter · Decision Letter 1]

13 Jun 2023

Dear Dr Strona,

We are pleased to inform you that your manuscript 'Time-travelling pathogens and their risk to ecological communities' has been provisionally accepted for publication in PLOS Computational Biology.

Best regards,

Rustom Antia

Academic Editor

PLOS Computational Biology

Natalia Komarova

Section Editor

PLOS Computational Biology

Reviewer's Responses to Questions

**Comments to the Authors:**

Reviewer #1: The manuscript has been greatly improved. In particular, the simulation program (Avidia) is now described early enough in the manuscript to fully understand the simulations and results. The authors have also added new analyses exploring potential confounding factors, reinforcing the robustness of the presented results. I have minor comments/questions left.

Please clarify how species are defined. It says in the text that they defined based on their genotype. What makes two individuals belong to two different species? One mutation in the code (would that be enough to talk about different species)? One different task (I would all call that a phenotype more than a genotype)?

Remove "catastrophic" line 63 (track-change version). If an emergence event does not succeed, I do not believe it would be catastrophic.

Switch "and" for "or" line 97.

Indicate that you use "host" and "free-living" for the same entity (right?) at the first occurence of "host" (e.g. line 98: "Aviadian (free-living (host) and...").

Do not start the new paragraph by "Another potential confounding effect", as this is the first time this notion is mentioned. Either state in the previous paragraph something like "A first potential confounding effect..." or the new one by "A potential confounding effect...".

Give a one sentence definition of Fermi estimations. Should by the way this paragraph go in the methods as a justification of the chosen numbers of simulation steps?

Reviewer #2: The authors have done a thorough job improving the paper in response to reviewer comments, thank you.

**Have the authors made all data and (if applicable) computational code underlying the findings in their manuscript fully available?**

Reviewer #1: Yes

Reviewer #2: Yes

PLOS authors have the option to publish the peer review history of their article (what does this mean?). If published, this will include your full peer review and any attached files.

Reviewer #1: No

Reviewer #2: No

---

## [Editor Report · Acceptance letter]

6 Jul 2023

PCOMPBIOL-D-22-01720R1 

Time-travelling pathogens and their risk to ecological communities

Dear Dr Strona,

I am pleased to inform you that your manuscript has been formally accepted for publication in PLOS Computational Biology. Your manuscript is now with our production department and you will be notified of the publication date in due course.

With kind regards,

Zsofi Zombor
